



# EnKF-based fusion of site-available machine learning air quality predictions from RFSML v1.0 and gridded chemical transport model forecasts from GEOS-Chem v13.1.0

Li Fang[1], Jianbing Jin*[1], Arjo Segers[2], Ke Li[1], Bufan Xu[1], Wei Han[3], Mijie Pang[1], Hai Xiang Lin[4,5], and Hong Liao*[1]

[1]Jiangsu Key Laboratory of Atmospheric Environment Monitoring and Pollution Control, Jiangsu Collaborative Innovation Center of Atmospheric Environment and Equipment Technology, School of Environmental Science and Engineering, Nanjing University of Information Science and Technology, Nanjing, Jiangsu, China
[2]TNO, Department of Climate, Air and Sustainability, Utrecht, The Netherlands
[3]Numerical Weather Prediction Center, Chinese Meteorological Administration, Beijing, China
[4]Institute of Environmental Sciences, Leiden University, Leiden, The Netherlands
[5]Delft Institute of Applied Mathematics, Delft University of Technology, Delft, the Netherlands

**Correspondence:** Jianbing Jin (jianbing.jin@nuist.edu.cn) and Hong Liao (hongliao@nuist.edu.cn)

**Abstract.** Statistical methods, particularly machine learning models, have gained significant popularity in air quality predictions. These prediction models are trained using the historical measurement datasets independently collected at the environmental monitoring stations, and their operational forecasts onward by the inputs of the real-time ambient pollutant observations. Therefore, these high-quality machine learning models only provide site-available predictions. In contrast, deterministic chemical transport models (CTM), which simulate the full life cycles of air pollutants, provide forecasts that are continuous in 3D field. However, owing to the complex error sources due to the emission, transport, and removal of pollutants, CTM forecasts are typically biased particularly in fine scale. In this study, we proposed a gridded prediction with high accuracy by fusing predictions from our recent regional-feature-selection machine learning prediction (RFSML v1.0) and a CTM forecast. The prediction fusion was conducted using the Bayesian theory-based ensemble Kalman filter (EnKF). Background error covariance was an essential part in the assimilation process. Ensemble CTM predictions driven by the perturbed emission inventories were initially used for representing their spatial covariance statistics, which could resolve the main part of the CTM error. In addition, a covariance inflation algorithm was designed to amplify the ensemble perturbations to account for other model errors next to the uncertainty in emission inputs. Model evaluation tests were conducted based on independent measurements. Our EnKF-based prediction fusion presented significant improvements than the pure CTM. Moreover, covariance inflation further enhanced the fused prediction particularly in the cases of severe underestimation.

## 1 Introduction

The rapid economic growth and urbanization has led to severe ambient air pollution in China (Li et al., 2016a). Thanks to the National Air Pollution Prevention and Control Action Plan released in 2013 (The State Council of China, 2013), the





atmospheric environment has been steadily improved. However, for the past decade, air pollution has been still ranked as the third major factor causing death in China, following tobacco and high blood pressure (Murray et al., 2020). Approximately 80% of the Chinese population is still exposed to primary fine particulate matter ($PM_{2.5}$) with annual mean concentrations exceeding 35 $\mu$g m$^{-3}$, and over 99% of the population is exposed to severe air pollution according to the World Health Organization air

quality guideline value of 10 $\mu$g m$^{-3}$. (Wang et al., 2019; Cheng et al., 2021b). Forecasting primary atmospheric pollutants with high spatial resolution is thus essential to provide early warning for residents and reduce detrimental public exposure (Bi et al., 2022).

Machine learning methods, particularly deep learning tools, have gained significant popularity in geoscientific fields owing to high accuracy and low computational resource requirements. A vast of studies have successfully applied machine

learning algorithms in predicting air quality. For example, Li et al. (2018) proposed a hybrid model combining weighted extreme learning machine and the adaptive neuro-fuzzy inference system for air quality predictions. Ma et al. (2020) improved the accuracy of WRF-Chem prediction of daily $PM_{2.5}$ concentrations in Shanghai by applying an XGBoost machine learning method. Cheng et al. (2021c) successfully predicted ground daily maximum 8-h ozone concentrations in typical cities of China by utilizing wavelet decomposition and two machine learning models. Mao et al. (2022) developed a dynamic graph

convolutional and sequence to sequence embedded with the attention mechanism model for predicting daily maximum 8-h average ozone concentrations. Recently, we successfully performed air quality prediction with high accuracy using the regional feature selection-based machine learning model (RFSML). Our forecast system is capable of providing national-scale short-term prediction over 1262 sites in China. Moreover, ensemble-SAGE feature selection algorithm was developed that can exclude those redundant inputs and efficiently improve the forecasting ability (Fang et al., 2022). These models were trained

via the historical measurement datasets collected at the air quality monitoring stations independently, and their operational forecasts forwards with the inputs of the real-time air quality observations. Unlike the gridded forecast, these predictions are only available over the location of the air quality monitoring sites. Meanwhile, the spatial distribution of existing environmental monitoring stations is rather uneven. For example, the monitoring network in China is dense in east and very sparse in west as shown in Fig. 1, and thus, the predictions over these limited sites can hardly represent the true $PM_{2.5}$ concentrations at national

scale (Liu et al., 2022).

Deterministic 3D chemical transport model (CTM) is the most popular tool used for operational air quality forecasting. Unlike machine learning-based air quality forecast models which typically rely on point-source observations, CTMs can predict air pollution in continuous spatiotemporal domain by modeling complex physical and chemical processes of air pollutant life cycles. Various CTMs have been developed and employed for air quality forecast. For example, Keller et al. (2021) provided a

new modeling system GEOS-CF that can make 5-day forecasts of the concentrations of five primary ambient pollutants; Cheng et al. (2021a) developed a real-time forecasting system of hourly $PM_{2.5}$ concentrations using WRF-CMAQ model in Taiwan. Lin et al. (2020) developed the WRF-GC model (coupling the Weather Research and Forecasting meteorological (WRF) and the GEOS-Chem model) that can perform high-resolution air pollutant forecasts. Using the WRF-Chem model, Georgiou et al. (2022) developed a high-resolution real-time air quality forecast system over the Eastern Mediterranean with better

performance than the Copernicus Atmosphere Monitoring Service. These CTMs could largely reproduce the spatiotemporal





variations in the ambient pollutants. However, owing to multi-source uncertainties in emission inventories (Keenan et al., 2009; Fan et al., 2018), initial and boundary conditions, and parameterization of physical and chemical processes e.g., transport and removal (Croft et al., 2012; Solazzo et al., 2017), the CTM forecast is susceptible to systematic bias particular at fine scale (Bi et al., 2022).

The same challenges exist when observations and simulation models are used to describe the atmospheric dynamics. Observations have the remarkable popularity owing to the higher accuracy than the numerical dispersion models. However, they are inevitably constrained in continuous 3D field and cannot provide complete scan of the full targeted domain. Models provide gridded simulating results which is typically biased as been explained before. To overcome the limitations of solely using models or observations, Bayesian theory-based assimilation methods (Evensen et al., 2022) by combining the observations

and model simulations have long been performed for obtaining gridded reanalysis that are considerably close to the reality. For example, the fifth generation of atmospheric reanalysis (ERA5) from ECMWF was produced by fusing atmospheric simulation from the Integrated Forecasting System (IFS) Cy41r2 and various types of measurements through a four dimensional variational (4DVar) data assimilation (Hersbach et al., 2020). To analyze desert dust aerosol along with its climatic interactions, Di Tomaso et al. (2022) developed a product with high resolution and continuous 3D field of dust aerosols over Northern Africa, the Middle

East, and Europe ranging from 2007 to 2016. The reanalysis was generated by assimilating (via local ensemble transform Kalman filter) MODIS AOD into their Multiscale Online Nonhydrostatic AtmospheRe CHemistry model (MONARCH).

In this study, the Bayesian theory based assimilation methods is innovatively introduced to fuse the machine learning forecast (RFSML v1.0) (Fang et al., 2022) and CTM air quality prediction. The fusion is designed for obtaining a gridded prediction with lesser bias and higher accuracy than the pure CTM prediction. Moreover, the prediction fusion is continuous

in 3D field unlike the machine learning forecast that are only site-available. To the best of our knowledge, this is the first time that assimilation method is applied in this way, which was generally applied for nudging model simulations with observations. The specific assimilation algorithm used is the ensemble Kalman filter (EnKF). Ensemble CTM forecasts are forwarded in parallel to representing the potential distribution of the ambient pollutant levels and the spatial covariance statistics, which are driven by perturbed emission inventories. To avoid model divergence, an extra covariance inflation algorithm is developed

that accounts for model errors other than the uncertainties in emission inputs. Uncertainty of the other prior, machine learning forecast, is also the essential part in assimilation fusion. To accurately quantify the errors, dynamic covariance is designed.

The remainder of the paper is organized as follows: Section 2.1 provides an introduction of the study domain and observations. A detailed description of EnKF assimilation methodology for fusing the machine learning and CTM forecasts is provided in Section 2.2. The machine learning forecast and CTM forecast are described in Section 2.3 and Section 2.4, respectively. A

popular spatial interpolation tool (cressman interpolation) that has the ability to expand the site-available machine learning forecast to a gridded one is illustrated in Section 2.5. In Section 3, independent evaluation of the proposed fused prediction is carried out. Conclusions and future prospects are provided in Section 4.





## 2 Data and methods

The modeling domain and time period used for testing our fused prediction are introduced in Section 2.1. The methodology of EnKF assimilation and localization used for fusing the machine learning and CTM forecast is illustrated in Section 2.2. The RFSML air quality forecast and its dynamic covariance are described in Section 2.3, and the CTM prediction and its spatial
covariance statistics are illustrated in Section 2.4. The Cressman spatial interpolation is explained in Section 2.5, which serves as a benchmark for comparison with our fused prediction.

### 2.1   Study domain and observations

The richness of the hourly measurements from air quality monitoring network established by the China Ministry of Environmental Protection (MEP) as shown in Fig. 1, enables the data-driven machine learning forecast at these stations. The
available sites are partitioned into six categories which is consistent with previous work (Fang et al., 2022), they are ones in the North China Plain (NCP; 34-41°N, 113-119°E), the Yangtze River Delta (YRD; 30-33°N, 119-122°E), the Pearl River Delta (PRD; 21.5-24°N, 112-115.5°E), the Sichuan Basin (SCB; 28.5-31.5°N, 103.5-107°E) and the Fenwei Plain (FWP; 33-35°N, 106.25-111.25°E; 35-37°N, 108.75-113.75°E). In this study, the proposed EnKF-based prediction fusion system was tested over the $PM_{2.5}$ concentration forecast over the entire China region. This method can be extended to other airborne pollutant
predictions in future studies. The winter of 2019 was selected as the test period following the choice in our recently work (Li et al., 2022) as winter suffers the most severe haze pollution than other seasons in China.

### 2.2   EnKF-based prediction fusion

The proposed assimilation-based prediction fusion is illustrated in Fig. 2. This figure shows the time series of hypothetical ambient pollutant predictions from both machine learning and pure CTM along the spatial coordinates, which could be X, Y,
or Z, without the loss of generality. The data-driven forecast using our RFSML system indicated by the blue line, provided accurate short-term forecast of the air pollutants that are very close to the reality (blue dots), as been validated in Fang et al. (2022) and can also be seen in Fig. 3. However, they are only available at limited sites where observation stations are located as explained before. Dynamic variance was introduced in this study to describe the uncertainty of the RFSML results as denoted by the light blue shadow. Unlike the data-driven forecast, the CTM model provides predictions over the continuous 3D filed
(indicated by gold curved surface), but it might contain unavoidable systematic bias. The Bayesian theory-based assimilation methodology is used to calculate the most likely posterior (or the fused prediction) given the potential spread of two priors.

The specific sequential assimilation system that is used to combine the site-available RFSML prediction and CTM prediction is the EnKF that was originally proposed by Evensen (1994) and further corrected by (Evensen, 2004). Similar to other assimilation algorithms, this assimilation system fundamentally relies on the Bayesian theory for finding the optimal posterior
that fits the two priors quantified by their covariance matrices.





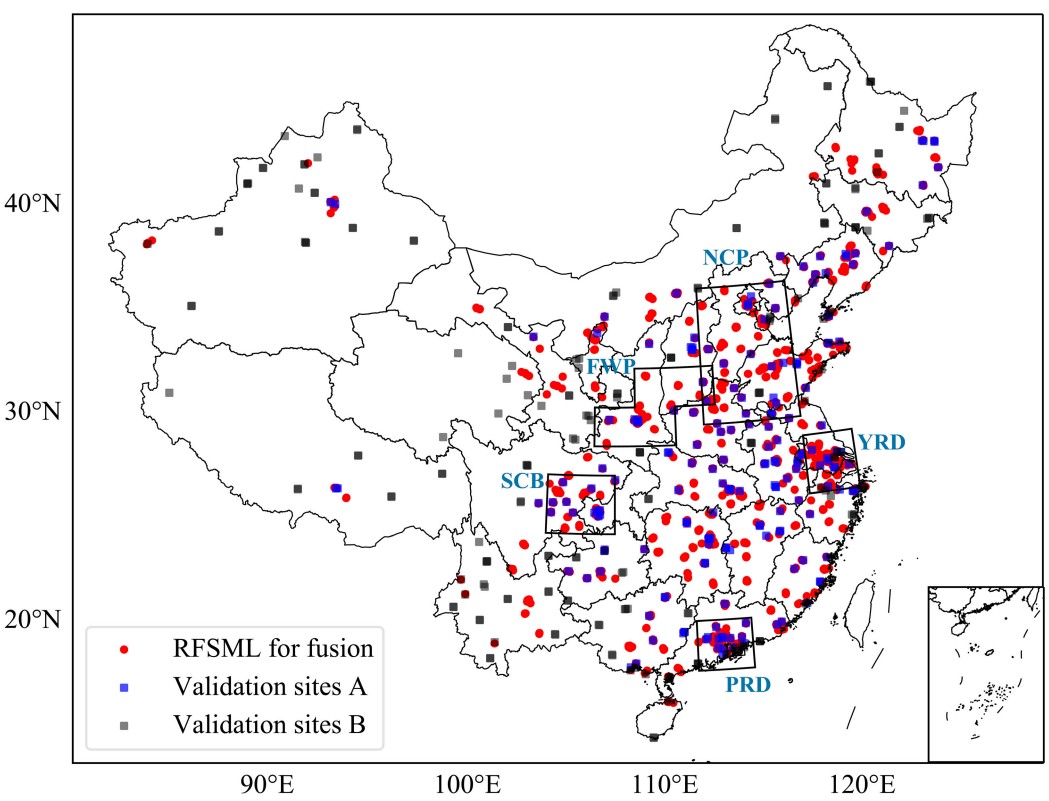

**Figure 1.** Distribution of air quality monitoring stations in China up to 2019. The black boxes represent the five main mega-city clusters classification used for the regional feature selection in RFSML. These RFSML prediction at sites (red dots) will be assimilated into the fused prediction. Observations from blue and black rectangles are used for independent evaluation.

To begin with, ensemble chemical transport model forecasts ($N$=32) are forwarded with perturbed emission inventories as will be discussed in Section 2.4 as follows:

$$[\boldsymbol{x}_1^f, ..., \boldsymbol{x}_N^f] \tag{1}$$

with $\bar{\boldsymbol{x}}^f \in \mathrm{R}^n$ equals the ensemble mean of $\boldsymbol{x}_i^f \in \mathrm{R}^n$, and $\mathbf{X}' \in \mathrm{R}^{n \times N}$ calculates the perturbation of the ensemble forecasts as:

$$\mathbf{X}' = [\boldsymbol{x}_1^f - \bar{\boldsymbol{x}}^f, \, ... \, , \boldsymbol{x}_N^f - \bar{\boldsymbol{x}}^f] \tag{2}$$

here $N$ represents the ensemble number while $n$ denotes the gridded chemical transport model size. The spatial background covariance matrix of the CTM prediction $\mathbf{P} \in \mathrm{R}^{n \times n}$ can be approximated using the ensemble perturbations via:

$$\mathbf{P} = \frac{1}{N-1} \mathbf{X}' \, \mathbf{X}'^{\mathrm{T}} \tag{3}$$



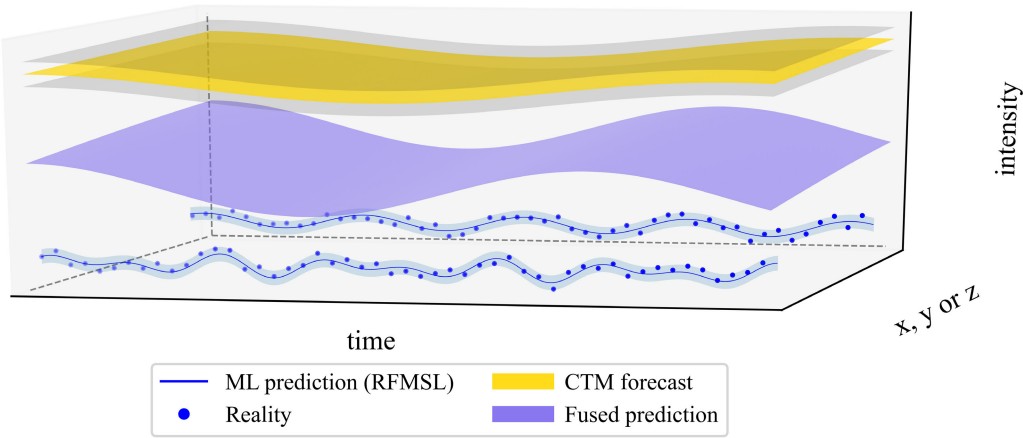

**Figure 2.** Framework of EnKF-based prediction fusion. The blue lines and their shadows imply for the RFSML prediction and its uncertainty at air quality monitoring stations, which are very close to the $PM_{2.5}$ reality. The gold surface and its surrounding grey surfaces stand for the CTM forecast and its uncertainty. The medium slate blue surface represents the fusion prediction of RFSML and CTM forecast.

The posterior forecast $x_a^f$ can then be fused according to the EnKF rules via:

$$x_a^f = \bar{x}^f + \mathbf{K}(\, y - \mathcal{H}\bar{x}^f\,) \tag{4}$$

where $y \in \mathrm{R}^m$ represents the RFSML machine learning forecast from $m$=1074 sites as will be explained in Section 2.3, $\mathcal{H} \in \mathrm{R}^{n \times m}$ is the linear operator that selects the gridded CTM prediction into the site-available machine learning forecast space,

5 and $\mathbf{K}$ denotes the Kalman gain, which can be calculated as follows:

$$\mathbf{K} = \mathbf{P}\mathcal{H}^{\mathrm{T}}(\, \mathcal{H}\mathbf{P}\mathcal{H}^{\mathrm{T}} + \mathbf{O}\,)^{-1} \tag{5}$$

where $\mathbf{O} \in \mathrm{R}^{m \times m}$ is the error covariance matrix of the machine learning forecast $y$ as will be illustrated in Section 2.3.

The classic EnKF has limitations such as its dependence on the relatively small ensemble number ($N$) compared to high model dimensions ($n$) to estimate background error covariance $\mathbf{P}$ dynamics (Houtekamer and Mitchell, 2001). To cut off those

10 spurious spatial correlation in $\mathbf{P}$, the most representative distance-dependent localization scheme (Lei and Anderson, 2014) is used. The localization is performed by multiplying a local support $\mathbf{L}$ via a Schur product as follows:

$$\mathbf{P}^{\mathrm{local}} = \mathbf{P} \circ \mathbf{L} \tag{6}$$

$$\mathbf{L}_{i,j} = \begin{cases} 1 - \frac{5}{3}\mathbf{S}_{i,j}^2 + \frac{5}{8}\mathbf{S}_{i,j}^3 + \frac{1}{2}\mathbf{S}_{i,j}^4 - \frac{1}{4}\mathbf{S}_{i,j}^5, & \mathbf{S}_{i,j} < 1 \\ -\frac{2}{3}\mathbf{S}_{i,j}^{-1} + 4 - 5\mathbf{S}_{i,j} + \frac{5}{3}\mathbf{S}_{i,j}^2 + \frac{5}{8}\mathbf{S}_{i,j}^3 - \frac{1}{2}\mathbf{S}_{i,j}^4 + \frac{1}{12}\mathbf{S}_{i,j}^5, & 1 \leq \mathbf{S}_{i,j} < 2 \\ 0, & \mathbf{S}_{i,j} \geq 2 \end{cases} \tag{7}$$





$$\mathbf{S}_{i,j} = \frac{\mathbf{D}_{i,j}}{L_{thres}} \tag{8}$$

here $\mathbf{D}_{i,j}$ represents the spatial distance between the grid cell $i$ and $j$, while $L_{thres}$ is the localization distance threshold. The individual elements of the local support $\mathbf{L}$ can be calculated using Eq. 8 and Eq. 7. The correlation $\mathbf{L}_{i,j}$ declines as the distance increases. The shorter distance threshold equals the greater descent rate. In this study, it was empirically set as 300 km which was tested to give the optimal performance.

## 2.3 RFSML prediction & uncertainty

Approximately 1500 air quality monitoring stations are present over China that provide hourly ambient pollutant measurements up to 2019 as shown in Fig. 1. Recently, the regional feature selection based machine learning forecast system (RFSML) was successfully developed for the short-term (with a horizon up to 24 h) air quality predictions. Given the regional key feature subset $\boldsymbol{a}_s = \{a_1, a_2, ..., a_s\}$, the RFSML can be described mathematically as follows:

$$\hat{y}_{t+h} = \mathcal{F}(a_1^{t-t_p+1}, \cdots, a_1^t, a_2^{t-t_p+1}, \cdots, a_2^t, \cdots, \cdots, a_s^{t-t_p+1}, \cdots, a_s^t) \tag{9}$$

where at any instant $t$, the input vector storing $s$ individual selected features over the previous $t_p$ hours is utilized to forecast the target $PM_{2.5}$ concentrations $\hat{y}$ with a prediction horizon of $h$ h. The highlights in RFSML was the ensemble-SAGE algorithm to select the regional key features, which resulted in remarkable improvements on the forecast effectively (Fang et al., 2022).

These high-quality predictions were available at 1262 stations (denoted as red dots and blue rectangles in Fig. 1), and 188 ones were skipped because of the high data missing rate in the data interpolation period (January 2018 to October 2019). Details concerning the strict data quality control can be found in Fang et al. (2022). These stations however still contain valuable measurements for validations. For this study, 188 stations that failed in the RFSML model training were used for validating our fused prediction, they are referred to as validation sites B and marked as black rectangles in Fig. 1.

Meanwhile, these 188 validation sites B are not evenly distributed over the entire modeling domain as can be seen in Fig. 1. To fully evaluate the forecasting ability of the proposed gridded prediction system, additional 188 sites were randomly selected from the 1262 RFSML stations and used for cross validations. They are referred to as validation sites A as shown in Fig. 1. Conclusively, the RFSML predictions at 1074 air quality monitoring stations (red dots) are used as one prior ($\boldsymbol{y}$) for the fused prediction, which are then compared with the measurements at 376 stations (blue rectangles and black rectangles) for validation, as can be seen in Fig. 1. Snapshots of our RFSML predictions at 1074 stations for assimilation are available in Fig. 6a-c, which exactly captured the spatial variations in the $PM_{2.5}$ as shown in Fig. S1 in the Supplementary Material.

As aforementioned, the error covariance matrix of the RFSML forecast ($\mathbf{O}$) is the essential input for Kalman gain calculation in Eq. 5. It governs the weight of the $\boldsymbol{y}$ prior in the optimization by describing its potential distribution. The errors in the RFSML predictions were assumed to be spatially independent, and hence $\mathbf{O}$ was diagonal. The RFSML errors not only varied in different stations, but also dynamically varied in a given site. The typical method is shown in Fig. 3(a), presenting the relationship between observations and RFSML prediction at a random site. The RFSML-predicted $PM_{2.5}$ values were relatively





close to the observations. Moreover, it presented high errors under severe-polluted scenarios. To explore the variations in the RFSML uncertainties, the samples shown in Fig. 3(a) were evenly divided into 10 collections (indicated by the gray dash line) based on the observation values. The mean values and root mean square errors (RMSE) of the 10 sample collections are plotted in Fig. 3(b), and their relationship was described using a linear function (blue solid line). Instead of characterizing the error

using a fixed value, the linear function was used to quantify the RFSML prediction error at the given station. The individual diagonal elements in $\mathbf{O}$ storing the square of the dynamic RFSML prediction error were then calculated by repeating the above calculation.

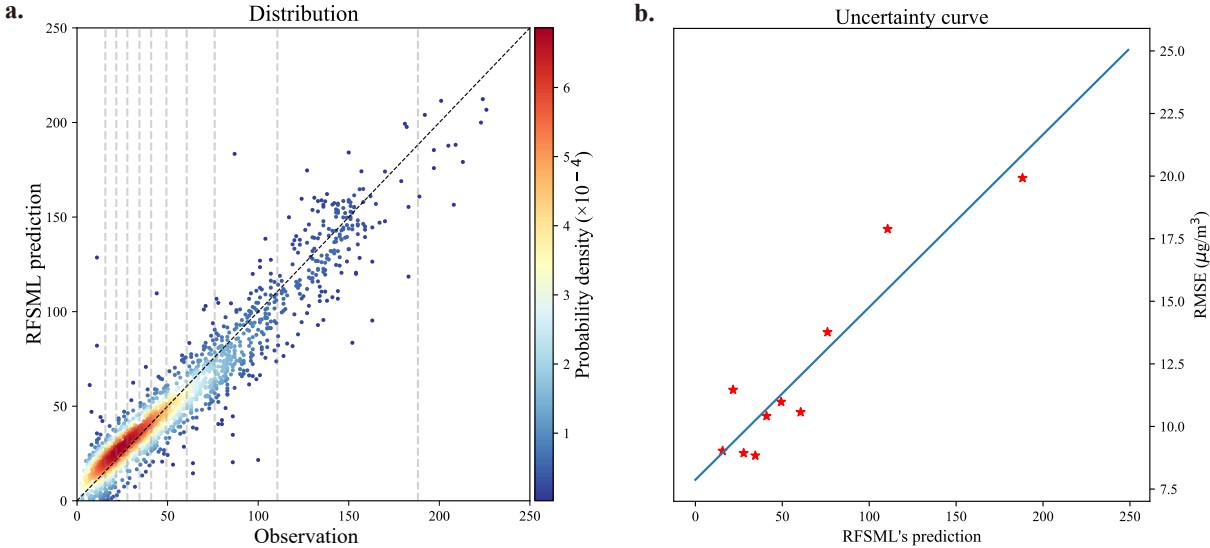

**Figure 3.** Dynamic uncertainty of the RFSML prediction at a random air quality station. Panel a is the distribution of the observation and RFSML prediction 6 h in advance. Panel b shows the linear fit of the average RFSML prediction and RMSE.

## 2.4    CTM forecast

The short-term CTM forecasts used for gridded prediction fusion in this study were from the GEOS-Chem v13.1.0 (DOI:

10.5281/zenodo.4984436) in a nested-grid simulation. It takes the global simulation with a horizontal resolution of 2° latitude by 2.5° longitude as the boundary condition. The nested domain for China modeling domain (70–140°E, 0–55°N) has a horizontal resolution of 0.5° latitude by 0.625° longitude and 47 vertical layers. This version had fully coupled aerosol-Ozone-NOx-hydrocarbon chemistry representation (Park et al., 2004; Dang and Liao, 2019). In this study, GEOS-Chem was driven by the archived Modern Era Retrospective analysis for Research and Applications, version 2 (MERRA-2) meteorological

fields (Gelaro et al., 2017). Notably, the reanalysis meteorology product was temporally used for testing our prediction fusion methodology. For the CTM prediction in practise, the operational meteorology forecast is essence, e.g., the GEOS-CF Keller et al. (2021) and WRF-GC system (Lin et al., 2020). The global anthropogenic emission inventory used in this study was





the Global anthropogenic emissions from the Community Emissions Data System (CEDS) inventory (Hoesly et al., 2018) which primarily contains aerosol, aerosol precursor, and reactive compounds. The monthly anthropogenic emission inventory for China is Multi-resolution Emission Inventory for China (MEIC, http://www.meicmodel.org) (Zheng et al., 2018). MEIC utilized here was the 2017 collection which is the latest version. Several natural emission sources were also included in the

model that can dynamically respond to the meteorological conditions, such as NOx emissions during lightning (Murray et al., 2012), and biogenic emissions which are computed online using MEGAN v2.1 (Guenther et al., 2012). To achieve successful operation of GEOS-Chem, a spin up of three month simulation was carried out before testing the 2019 winter PM$_{2.5}$ forecast. The PM$_{2.5}$ concentrations were calculated as the sum of the concentrations of the sulfate, nitrate, ammonium, black carbon, and organic carbon in this study.

**2.4.1 CTM forecast covariance**

The uncertainty in the GEOS-Chem forecast is initially attributed to the errors in the emission inventories. It is assumed to be compensated using a spatially varying tuning factor similar to the approach in related work (Di Tomaso et al., 2017; Jin et al., 2018) as follows:

$$\boldsymbol{f}^{\text{true}}(i) = \boldsymbol{f}_b(i) \cdot \boldsymbol{\alpha}(i) \tag{10}$$

where $\boldsymbol{f}_b(i)$ denotes the aerosol emission rate in the given grid cell $i$ from the MEIC, and $\boldsymbol{f}^{\text{true}}(i)$ represents the true value. The $\alpha$ values are defined to be random variables with a mean of 1.0 and a standard deviation $\sigma_\alpha$ = 0.2. This empirical value was found to provide sufficient freedom for resolving the observation-minus-simulation errors to a large extent. A background covariance $\mathbf{B}_\alpha$ was formulated as a product of the constant standard deviation and a spatial correlation matrix $\mathbf{C}$:

$$\mathbf{B}_\alpha(i,j) = \sigma_\alpha \cdot \mathbf{C}(i,j) \tag{11}$$

where $\mathbf{C}(i,j)$ represents a distance-based spatial correlation between two $\alpha$s in the grid cell $i$ and $j$, and is defined as:

$$\mathbf{C}(i,j) = e^{-(d_{i,j}/l)^2/2} \tag{12}$$

where $d_{i,j}$ represents the distance between two grid cells $i$ and $j$. $l$ here denotes the correlation length scale which controls the spatially variability freedom of the $\alpha$s. A small $l$ means more errors in fine scale could be resolved using the assimilation, while however requires more ensemble runs to represent the model realization from emission to simulation as will be explained later.

An empirical parameter $l$ = 300 km used in EnKF to cut off the dust emission that has a rapid spatially variability was also considered in this study.

With $\mathbf{B}_\alpha$ that describing the potential spread of the true emission situation, ensemble emission inventory $[\boldsymbol{f}_1,...,\boldsymbol{f}_N]$ could then be generated randomly. They will then be used to onward our GEOS-Chem model $\mathcal{M}$ for ensemble PM$_{2.5}$ forecasts in Eq. 1 via:

$$[\boldsymbol{x}_1^f, ..., \boldsymbol{x}_N^f] = [\mathcal{M}(\boldsymbol{f}_1), ..., \mathcal{M}(\boldsymbol{f}_N)] \tag{13}$$





### 2.4.2 Covariance inflation

The perturbed emission inventories could resolve the deficiency in the model forecast effectively as will be discussed in Section 3 by feeding the site-available RFSML result. However, the posterior forecast error occasionally remained at high values especially when the prior CTM severely underestimated the pollution levels. This can be best seen in Fig. 4. Panel a

shows the times series plot of prediction from RFSML with the uncertainty, mean CTM forecasts with its uncertainty, $PM_{2.5}$ concentration measurements, and fused result at a station labeled as 1063A. Note that that mean and uncertainty (standard deviation) of the CTM are calculated based the ensemble CTM simulations, which could be found in Fig. S2(a) as well. The difference between the fused prediction (sky-blue line) and observations (red stars) is not resolved steadily, especially in the red marked regions. The uncertainty quantified by the CTM spread (silver shadow) are far less than the RFSML uncertainty

(blue shadow). Therefore, the prior of CTM prediction weighted much higher than the prior of RFSML in the assimilation. This resulted in a posterior prediction at a level similar to that of the mean of the ensemble forecast (black dot), and deviated considerably from the RFSML prediction (blue line). This could be attributed to the fact that the perturbed MEIC emission uncertainty only partially accounts for the simulation-minus-observation error in CTM prediction. However, the unconsidered error caused by meteorology, deposition and other process could also contribute to the simulation-minus-observation error.

To compensate for these inevitable errors in the CTM uncertainty and avoid the assimilation divergence, covariance inflation was designed. The basic idea was to amplify the ensemble perturbations while maintaining the mean via:

$$\boldsymbol{x}_i^{\text{inflate}}(j) \;=\; \bar{\boldsymbol{x}}(j) + \beta[\, \boldsymbol{x}_i(j) \,-\, \bar{\boldsymbol{x}}(j)\,] \tag{14}$$

where $\boldsymbol{x}_i(j)$ represents the original prediction from the ensemble $i$ at grid cell $j$ while $\boldsymbol{x}_i^{\text{inflate}}$ denotes the inflated one, and $\beta$ is the inflation factor for amplifying the ensemble perturbation with respect to the ensemble mean $\bar{\boldsymbol{x}}(j)$, which is defined as

follows:

$$\beta = 15 \cdot e^{-(\bar{\boldsymbol{x}}(j)/5)} + 1.5 \tag{15}$$

As the ensemble mean increase, the inflation factor declines smoothly from the a maximum of 16.5 to a minimum of 1.5 as shown in Fig. 4(c). The spread of the resampled ensemble CTMs is available in Supplement Fig. S2(b). Higher inflation was set for these low-value predictions to compensate for the uncertainty raised from meteorology or other transport processes. The

posterior prediction could subsequently be calculated with the inflated covariance.

Fig. 4(b) shows the time series of the uncertainty of the resampled ensemble members, presenting a much wider spread compared to the original ones in panel a. This effectively avoids assimilation divergence and allows the posterior prediction to be nudged toward RFSML. Overall improvement on gridded prediction against independent measurements will be discussed in Section 3.

### 2.5 Spatial interpolation benchmark: Cressman interpolation


Interpolating data from observational stations to regular grid cells is also a hot topic in the geoscience (Yu et al., 2011). Many deterministic and geospatial tools for spatial interpolation have been developed, such as the Cressman interpolation

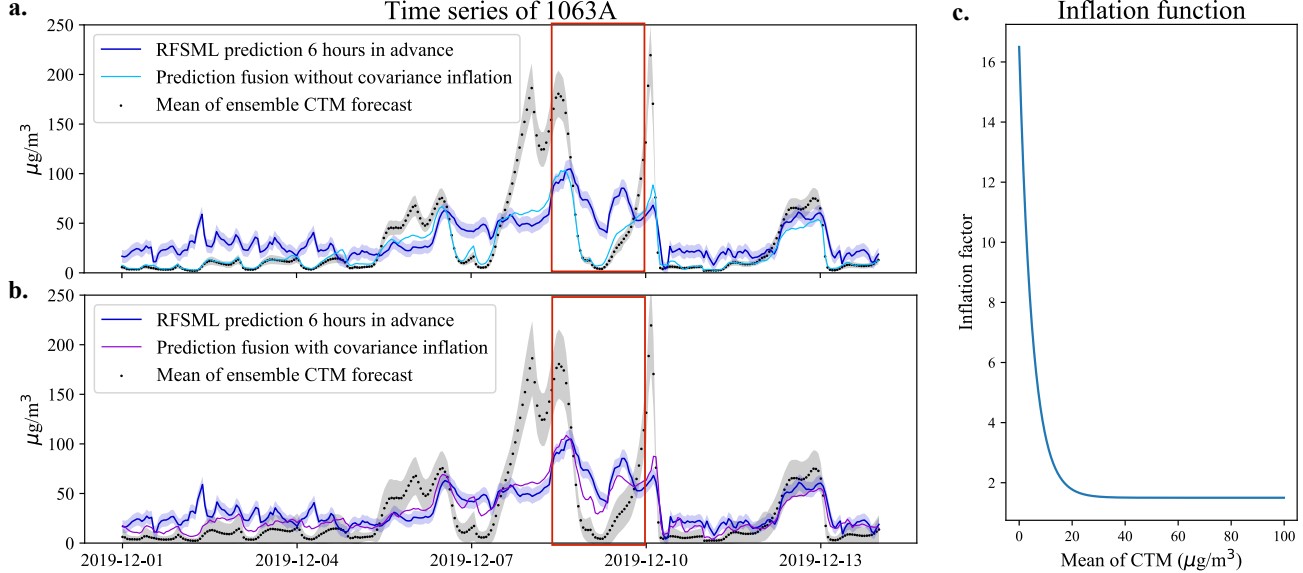

**Figure 4.** Panels a and b are the time series of an environmental monitoring station (Latitude:40.98°N, Longitude:117.95°E) in Chengde, Hebei Province. RFMLS prediction is available at this station and will be assimilated. Markers of medium blue solid line, deep sky-blue solid line, dark violet solid line, red star, and black dot represent the RFSML prediction, prediction fusion without covariance inflation, prediction fusion with covariance inflation, ground observation, and mean of ensemble CTM predictions respectively. The silver and blue shadow represent the uncertainty of the CTM and RFSML respectively. Panel c is the inflation function.

(Cressman, 1959), the Kriging interpolation (Oliver and Webster, 1990; Stein, 1999), the inverse distance weighting (Bartier and Keller, 1996), and etc.. Those methods can transfer the site-available RFSML prediction into the continuous 3D field product alternatively. All these methods are based on the assumption that the weight is inversely proportional to the distance between the predicted location and the sampling location (Lu and Wong, 2008). Therefore, they are therefore relatively simple

5 and efficient in computation compared to our proposed prediction fusion method, which relies on the ensemble CTM models to represent the spatial covariance statistics.

In this work, these popular tools including the Ordinary Kriging interpolation, Cressman interpolation and inverse distance weighting were also tested to interpolate our site-available RFSML prediction for obtaining a gridded one. They are served as the benchmark for comparing against our proposed assimilation-based fusion method. To ensure a fair comparison,

10 the statistical interpolation methods were performed based on the RFSML prediction at 1074 sites which were used for our assimilation-based fusion. However they failed to forecast the spatial pattern of the $PM_{2.5}$ concentration either in the national scale or in fine scale. Of the three, the Cressman interpolation provided the most optimal results. Snapshots of the prediction interpolation are shown in panels a, b, and c in Fig. 5 with a scaling radius of 3°, 5°and 10°, respectively. The typical limitations of using these distance-weighted methods can be clearly observed from the figure. While parts of the $PM_{2.5}$ spatial pattern





were captured with the hottest spot in the North China plain using the smallest search radius in panel a, it failed to obtain the full gridded prediction as the air quality monitoring stations are sparsely distributed especially in the western areas. However, when using a large scaling radius, most of the spatial dynamics are lost with a huge discrepancy against the independent measurements (indicated by colored circles) in panel c. Therefore, distance-weighted methods cannot satisfy the motivation for

obtaining a 3D continuous prediction starting from the site-available machine learning forecast.

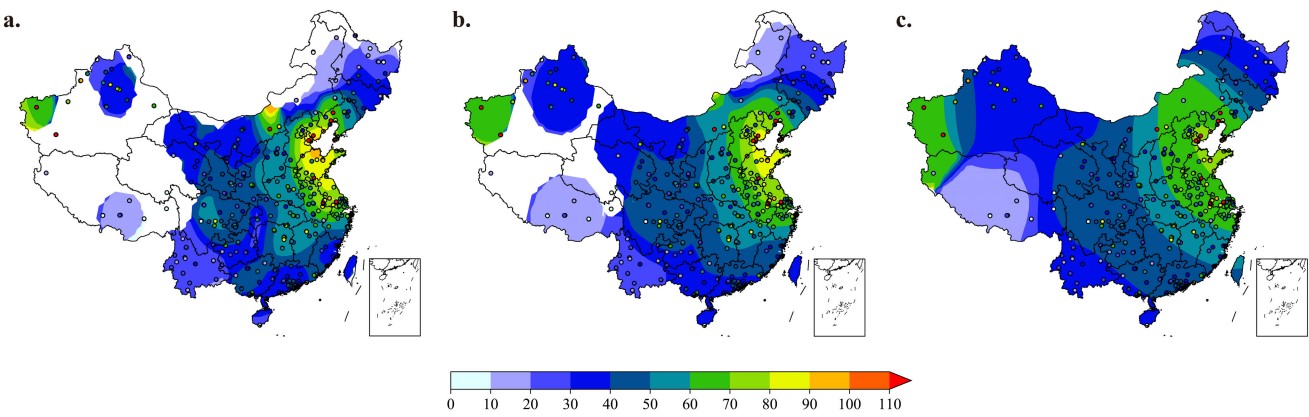

**Figure 5.** Snapshots of the distribution of the PM$_{2.5}$ concentration forecasts and observations at at 2019/10/30 16:00:00 (UTC). Panel a, b and c represent Cressman interpolation of the RFSML (6 h in advance) predictions with a scaling radius of 3°, 5°and 10°. The colored circles imply the independent observations from 376 ground stations which are not used as source of interpolation.

## 3   Results and Discussion

The forecast skills of our gridded prediction through fusing machine learning and CTM prediction using EnKF was fully evaluated. The performance of the pure CTM forecast is first discussed in Section 3.1. Section 3.2 illustrates the overwhelming skill of the proposed prediction in terms of the time series prediction at single stations, and the necessity of using covariance

inflation is emphasized. Overall performance in the spatial pattern of the fused prediction against the independent observations is further evaluated over the whole test period in Section 3.3. The validation of the CTM and evaluation of fused prediction performance was primarily conducted using RMSE, mean absolute error (MAE) and Pearson correlation coefficient (R). The formulas determining these metrics are listed in supplement Formula (2-4).

### 3.1   Pure CTM forecast

An ensemble of 32 CTM predictions with disturbed MEIC emission inventories were forwarded to quantify the spatial covariance statistic of the PM$_{2.5}$ concentration forecast as discussed in Section 2.4. In addition, a base run driven by the default MEIC emission inventory was performed over the test period, 2019 winter, to verify the forecast skill of the pure CTM. It was considered as the benchmark for validating our fused prediction.





The overall evaluation results of the CTM $PM_{2.5}$ forecast in terms of RMSE and MAE over the test period are shown in Fig. 8. Though the CTM can reproduce the $PM_{2.5}$ spatial and temporal variation to a larger extent as will be discussed in Section 3.2 and Section 3.3 in details, a difference in $PM_{2.5}$ intensity existed obviously. This resulted in relatively high RMSE and MAE; in particular, the RMSE and MAE were as high as 42.8 and 29.7 $\mu g\ m^{-3}$ and 47.0 and 32.8 $\mu g\ m^{-3}$ in severely

polluted regions such as NCP and FWP, respectively. Moreover, significant overestimation in the SCB region accounts for high RMSE and MAE (53.1 and 42.6 $\mu g\ m^{-3}$) which is consistent with Li et al. (2016b). More model validation of the CTM is shown in Supplement Fig. S3 with a normalized mean bias (NMB) of -6.87% over the entire test period. Details concerning improvements of the proposed fused prediction over the pure CTM will be given in Section 3.3.

### 3.2    Time series of single monitoring station

One of the 376 environmental monitoring stations for independent validation, labeled as 1812A (34.65°N, 112.39°E), was selected as the typical example for the time series discussion. It was used to illustrate the typical results that were observed in other validating sites. Panels a and b in Fig. 6 show the pure CTM and fused forecasts (6 and 18 h in advance, respectively) against the independent $PM_{2.5}$. Though the CTM captured the temporal dynamics in 1812A in general, the difference in the magnitude was sometimes quite large. Through assimilating the spatial pattern from the site-available RFSML result, the fused

gridded prediction outperforms the purely CTM forecast obviously. Notably, the optimal result is obtained when covariance inflation is implemented. The posterior prediction with covariance inflation performed better, especially in those low-value $PM_{2.5}$ situations as has been explained in Section 2.4.2.

The benefit of the covariance inflation is highlighted when the RFSML forecast in longer prediction horizon are fused as shown in panel b (18 h). This is because the RFSML dynamic error grows steadily as the prediction length increases (Fang

et al., 2022). Without using the covariance inflation, the assimilation algorithm would trust the CTM over RFSML. It would then lead to a result that stays closer to the CTM prior. A similar outcome can be observed in supplement Fig. S4 which is the time series diagram of the same station with forecast horizons of 12 and 24 h. There is one exception that is around 2019/Dec/23 (UTC), where CTM overestimates the $PM_{2.5}$ concentrations while prediction fusion all underestimates it. This is mainly due to the abnormally low prediction values from the RFSML at nearby sites. In general, the proposed prediction

fusion exhibits significant advantages over CTM, and the adopted covariance localization effectively prevents the assimilation divergence and further improves the gridded prediction fusion.

### 3.3    Spatial forecast

Fig. 7 shows the spatial distribution of the fused predictions (panels g-i) and the ones using extra covariance inflation (panels j-l) against the independent $PM_{2.5}$ measurements at three randomly selected tested instants. Those two gridded posteri-

ors were obtained via fusing the site-available RFSML in panels a-c and CTM prediction in panels d-f. Stead improvements on the spatial pattern are observed in our fused prediction compared to the pure CTM. For instance, the CTM underestimated the $PM_{2.5}$ pollution over NCP region at the first (red box in panel d) and third instants (red box in panel f). The underestimation was partially relieved by assimilating the RFSML prediction (see panels g and i). The CTM's overestimations in SCB (panel

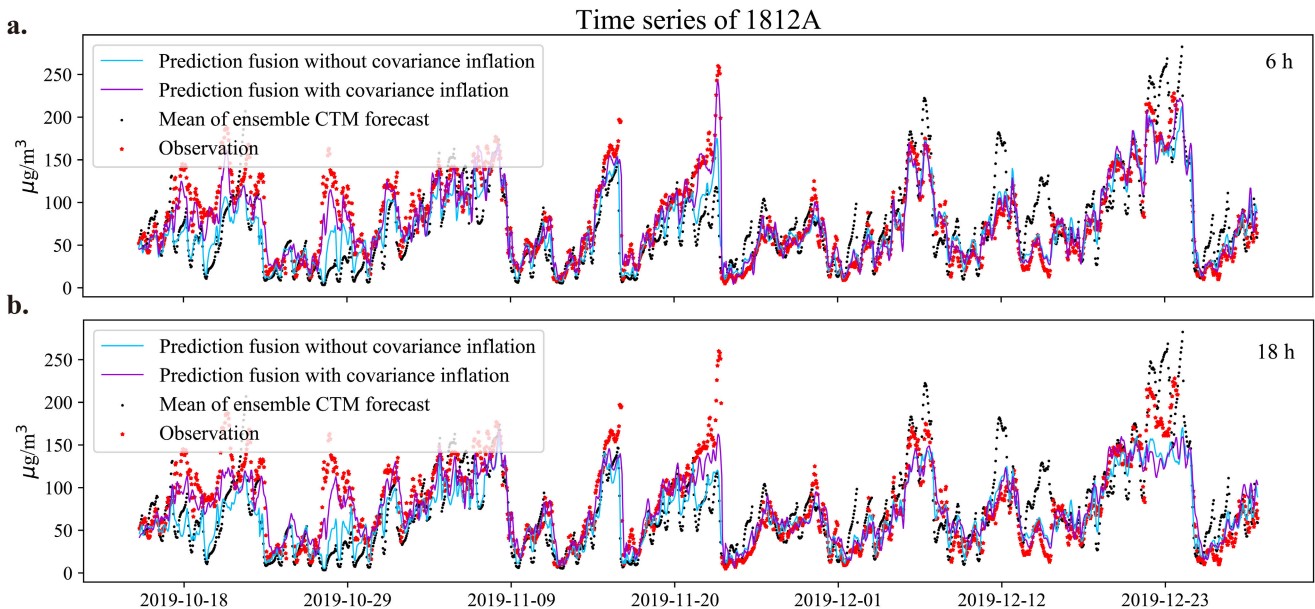

**Figure 6.** Time series of an environmental monitoring station (Latitude:34.65°N, Longitude:112.39°E) in Luoyang, Henan Province. This station is one of the validation set. Markers of deep sky-blue solid line, dark violet solid line, red star, and black dot represent prediction fusion without covariance inflation, prediction fusion with covariance inflation, ground observation, and mean of ensemble CTM predictions respectively. Panels a and b represent forecasts 6 and 18 h ahead, respectively.

e) and the south of China (panel f) were also relieved to a great extent by the proposed prediction fusion (see panel h and i). The predictions presented in panels j, k, and l are in better harmony with the measurements when covariance inflation is implemented in these examples above. When severe CTM underestimation occurred in the North China (green box in panels) especially the Xinjiang province, solely assimilating the RFSML prediction was not sufficient to correct it. However, the application of covariance inflation could effectively resolve the underestimation as shown in panels j, k, and l (green box).

In summary, the fused prediction obtained through assimilating the high-quality RFSML prediction could effectively improve CTM spatial variability prediction. Additionally, the covariance inflation can further enhance the performance of the prediction fusion especially in places with severe underestimation. This is mainly because the perturbed MEIC emission only partially accounts for the simulation-minus-observation error. The error caused by meteorology, deposition, and other processes should also be taken into the account as has been done in our proposed covariance inflation.

The overall performance of our EnKF-based prediction fusion with or without covariance inflation over the whole testing period were evaluated. As shown in Fig. 8, the proposed prediction fusion has much lower RMSE and MAE than that of CTM in most validation stations. There are a few sites where the improvements are limited, for instance in Xinjiang Province. This is mainly because there are too few RFSML sites nearby for assimilation. The fused prediction with covariance inflation further



**Figure 7.** Snapshots of PM$_{2.5}$ forecast 6 h in advance at three instants (each column indicates the same moment). Panels a-c show the RFSML predictions (colored dots) at 1074 air quality stations. Panels d-f show the CTM forecast and the ground observation (colored dots). Panels g-l show the prediction fusion (without and with covariance inflation) results and the reality (colored squares) from 376 evaluation stations.



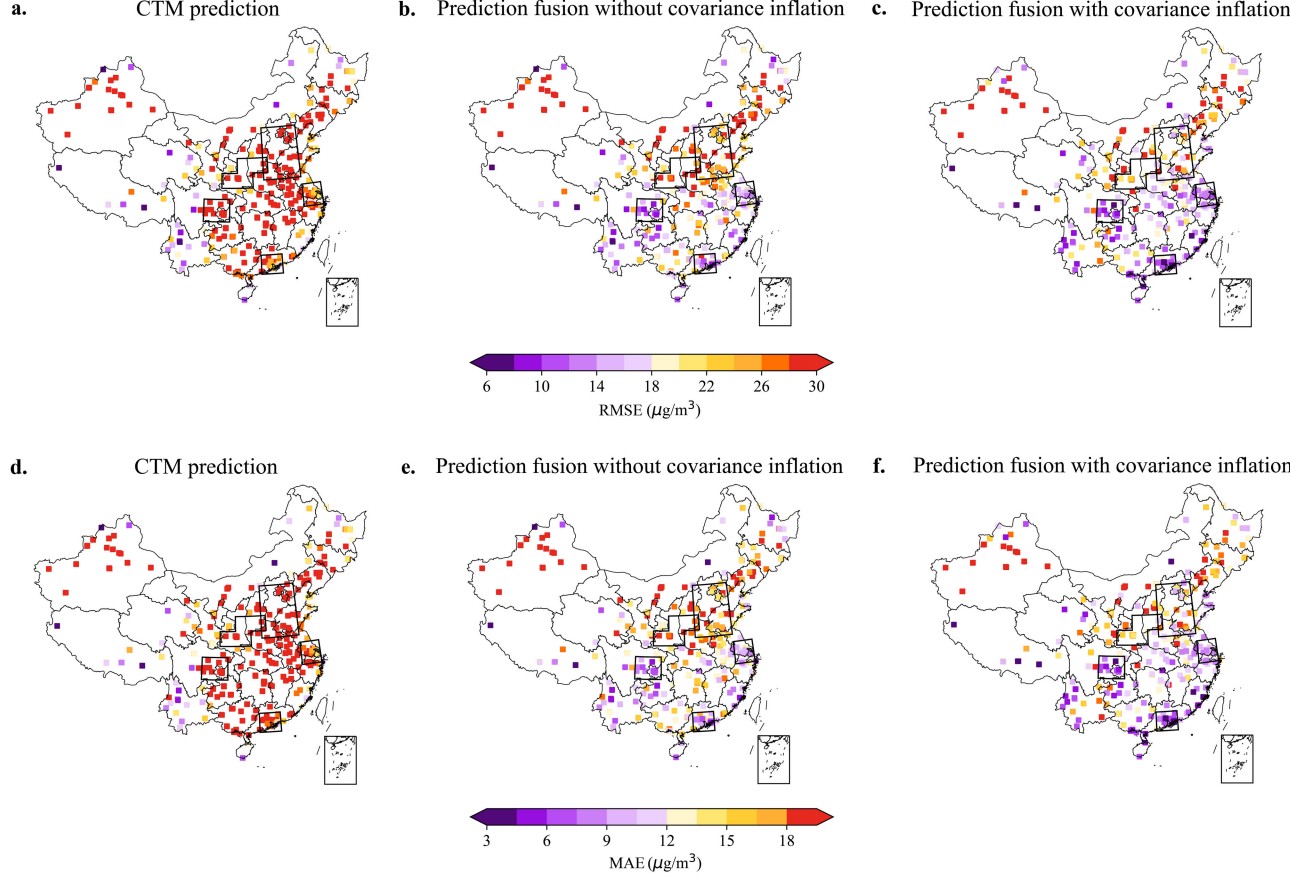

**Figure 8.** Spatial statistics of CTM, prediction fusion without covariance inflation and prediction fusion with covariance inflation. Panels a-c and panels d-f represent statistic results of RMSE and MAE respectively. The results are based on 6 h in advance prediction.

enhanced the prediction skill, especially in places with sufficient RFSML sites for assimilation which are mainly located in the five main mega-city clusters (black boxes).

To clearly visualize the benefit of using the EnKF-based prediction fusion method and covariance inflation, a modified Taylor diagram (Taylor, 2005) is given in Fig. 9, which shows the RMSE and R of the CTM and our fused prediction over the five clusters simultaneously. These RMSE and R are calculated with respects to the regional independent air quality monitoring sites.

The advantage of using our prediction fusion method against the pure CTM prediction is consistently visible for all the five regions and various predicting horizons (6, 12, 18, and 24 h). Take the panel a for instance, the CTM has the worst prediction in terms of R (<0.1) in PRD region; however, it increases up to 0.62 when the RFSML forecast is assimilated, and further increases to 0.85 when covariance inflation is implemented. In terms of the RMSE, the most remarkable improvement is obtained in SCB region. Our EnKF-based fusion reduced the RMSE from 43 $\mu$g m$^{-3}$ to 12.73 $\mu$g m$^{-3}$ and 10.97 $\mu$g m$^{-3}$ (with covariance





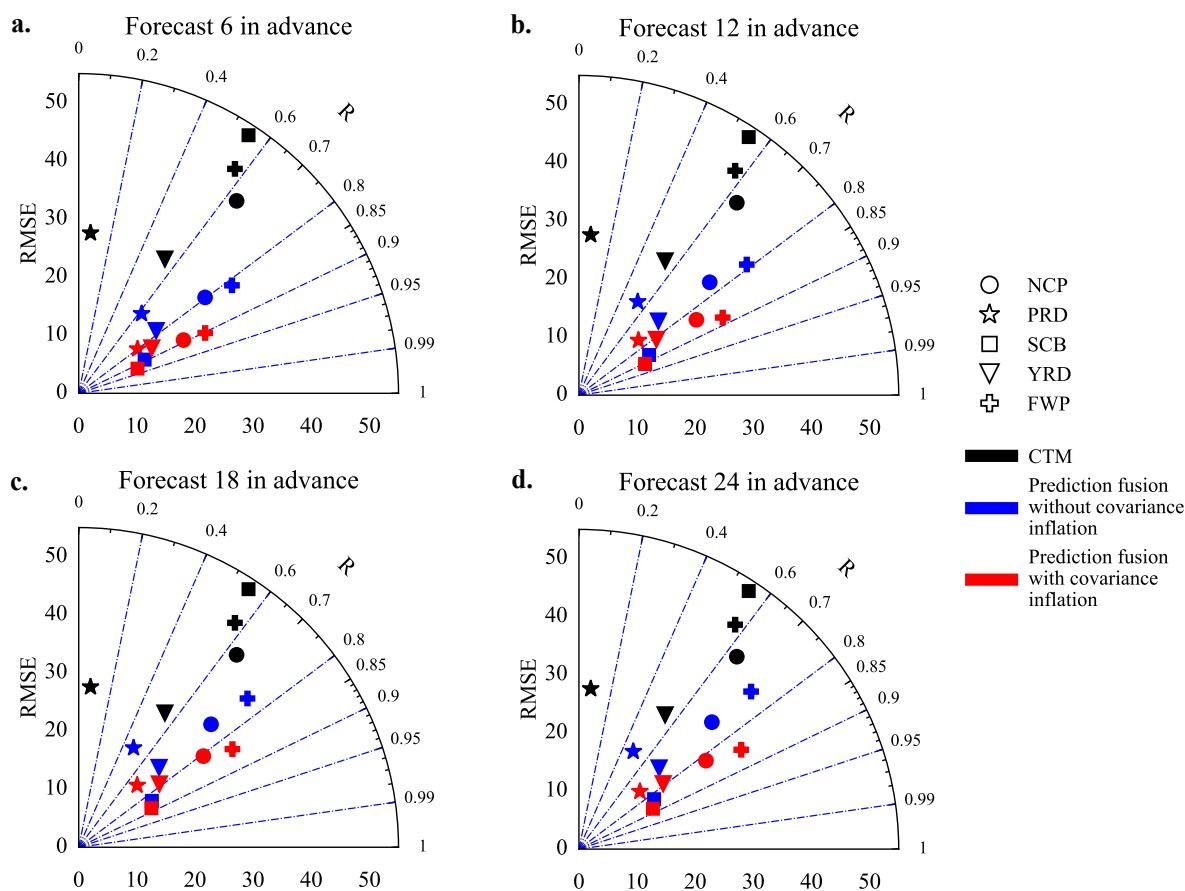

**Figure 9.** A modified Taylor Diagram that illustrates RMSE and R together. NCP, PRD, SCB, YRD and FWP were represented by different markers of circle, star, square, triangle_down, plus and diamond respectively. Colors of black, blue, and red indicate results from CTM, prediction fusion without covariance inflation and prediction fusion with covariance inflation respectively. The panels from top left to bottom right are forecasts 6, 12, 18, and 24 h ahead.





inflation). This is a considerable improvement and can be attributed to the densely distributed RFSML prediction sites in the region as can be seen in Fig. 1. Note that our fused prediction skill also generally declines with an increase in prediction length following the RFSML. Therefore, prediction fusion with longer forecasting horizon was not carried out. Thus, the prediction fusion method has much better prediction performance than CTM and covariance inflation can further enhance this advantage

in all the tested predicting horizons.

## 4  Conclusion

Machine learning models have strong advantages when applied to air quality predictions. However, their high-quality predictions are only available in air quality monitoring stations. CTM can forecast ambient pollutants in a time and spatial continuous manner, but the accuracy is not guaranteed as it is limited by various error sources, such as emission inventory,

meteorology, and initial and boundary conditions among others. In this study, we proposed a EnKF based method to fuse the site-available machine learning prediction (RFSML v1.0 in this study) and CTM forecast. In the assimilation calculation, the uncertainty of RFSML results is quantified by a dynamic covariance while the uncertainty of the CTM prediction are represented using ensemble realizations driven by perturbed emission inventories. It results in a relatively accurate prediction that is continuous in 3D field as well. The proposed prediction fusion presented remarkable performance over the pure CTM

as indicated by metrics include Pearson correlation (R), MAE and RMSE. Take the prediction with 6 h horizon in the five mega-city clusters for instance, the average RMSE was reduced from 48.82, 27.66, 53.14, 27.42, and 47.04 $\mu$g m$^{-3}$ to 27.28, 17.46, 12.73, 17.11, and 32.24 $\mu$g m$^{-3}$ in NCP, PRD, SCB, YRD and FWP respectively. The corresponding R increased from 0.63. 0.07. 0.55, 0.54, and 0.57 to 0.79. 0.62, 0.89. 0.78, 0.82, and 0.68 simultaneously.

Meanwhile, for the CTM, there are unconsidered but non-negligible uncertainties (meteorology, deposition, and initial

and boundary conditions) in addition to the uncertainty in emission inventory which was the only one considered initially. These uncertainties of CTM were taken into accounted as well through the application of covariance inflation. It helps to fully represent the CTM errors and avoid the assimilation divergence, further enhancing its forecast skills. Through re-weighting the two priors using the empirical covariance inflation, predicting fusion achieves the best posterior prediction results. Specifically, it successfully reported some local severe pollution such as in Xinjiang Province, and captured fine-scale PM$_{2.5}$ variation in

regions with complex pollution patterns. Notably, the average RMSE of the PM$_{2.5}$ prediction in those five densely-populated clusters (NCP, PRD, SCB, YRD, and FWP) was further reduced to 20.22, 12,68, 10.97, 14,78, and 24.10 $\mu$g m$^{-3}$ respectively. The corresponding R with the application of covariance inflation was further increased to 0.89, 0.80, 0.92, 0.85, and 0.90 effectively.

## Code and data availability

The ground-based air quality monitoring observations are from the network established by the China Ministry of Environmental Protection and accessible via https://quotsoft.net/air/. The GEOS-Chem v13.1.0 source code is archived on Zen-





odo (https://zenodo.org/record/4984436) and MEIC inventory for modeling the anthropogenic activity emission can be obtained via http://www.meicmodel.org/. The PM$_{2.5}$ data used in this paper and the model output data are archived on Zenodo (https://doi.org/10.5281/zenodo.7619183); Li Fang, 2023). The python source code of EnKF-based prediction fusion is archived on Zenodo (https://doi.org/10.5281/zenodo.7439497; Li Fang, 2022).

## 5 Acknowledgments

This work is supported by the National Natural Science Foundation of China [grant nos. 42105109 and 42021004] and Natural Science Foundation of Jiangsu Province (grant nos.BK20210664 and BK20220031).

## Author contribution

JJ conceived the study and designed prediction fusion method. JJ and LF wrote the code of the prediction fusion. LF carried out the prediction and evaluation. AS, KL, BX, WH, MP HXL and HL provided useful comments on the paper. LF and JJ prepared the manuscript with contributions from all others co-authors.

## Competing interests

The authors declare that they have no conflict of interest.





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
