# Peer review of "A gridded air quality forecast through fusing site-available machine learning predictions from RFSML v1.0 and chemical transport model results from GEOS-Chem v13.1.0 using EnKF"

_Geoscientific Model Development, 2022_

## Author Comment (AC1)

**Response to Referee #1:** We would like to thank the referee for the careful review throughout the paper that help to improve our paper.

Our Reply follows (*the editor's comments are in italics and blue*)

*General Comments*

*In this article, the author proposes a high-precision grid prediction by fusing regional-feature selection machine learning prediction (RFSML v1.0) and CTM prediction. The set CTM prediction driven by the disturbance emission inventory is used to represent its spatial covariance statistics to solve the error, and the covariance expansion algorithm is designed to amplify the integrated disturbance and reduce the error, and the model evaluation is carried out on the basis of independent measurement. The prediction fusion based on EnKF is significantly improved than pure CTM. The manuscript has a good innovation, the content of the manuscript is detailed, the discussion is explicit, and the conclusion is clear. It is suggested to add some content for minor revision and publish it in the journal Geoscientific Model Development.*

*Comments*

*What are the input variables of machine learning model (RFSML)?*

**Reply:** Thanks for the comments and this was indeed not clearly explained in our previous version. In our prior work, we identified the top three significant features for each region, as outlined in Table 6 of our publication. Additionally, we utilized the preceding nine hours for the forecast, resulting in a 27-dimensional input matrix. To explain this, remarks are now added in page 4, line 14-16 "***We identified the top three significant features for each region, as outlined in Table 6 of our publication (Fang et al., 2022). Additionally, we utilized the preceding nine hours for the forecast, resulting in a 27-dimensional input matrix for models.***"

*It is suggested to add time-based verification results, that is, to use the data of two months for modeling and the data of the third month for verification.*

**Reply:** Time-based verification is crucial for machine learning. In RFSML, we utilized 15,690 hours (from January 1, 2018, to October 15, 2019) for model training and cross-validation. The remaining 1824 hours of data from October 15 to December 30, 2019, were reserved for actual testing.

To further verify the predictive capability of RFSML in a rolling fashion, we tested the forecast fusion for a less polluted month, April 2020. The details could be found in the response to Question "*What is the effect if the model is applied to other seasons?*" in the below.

*It is suggested to use a schematic diagram to show the variable input and output relationship between the machine learning model and the chemical transfer model, so as to facilitate readers' understanding.*

**Reply:** We agree with the referee that including the input relationships would make the picture more comprehensible. We added the two inputs of EnKF in Figure 2 as shown below.

[Figure]

*Figure 2. Framework of EnKF-based prediction fusion. The blue lines and their shadows imply for the RFSML prediction and its uncertainty at air quality monitoring stations, which are very close to the $PM_{2.5}$ reality. The gold surface and its surrounding grey surfaces stand for the CTM forecast and its uncertainty. The medium slate blue surface represents the fusion prediction of RFSML and CTM forecast.*

*In what aspects is the feature selection of machine learning region reflected?*

**Reply:** In our previous study, we demonstrated the significant role of feature selection in machine learning and conducted it for different regions. As the impact of feature selection varies across regions, the prediction performance of machine learning also varies across regions. Therefore, the use of prediction results from previous work affected the prediction performance in different regions in our study. Furthermore, the prediction performance of the CTM forecast varied across regions and also had an impact on our method.

*What is the maximum forecast time for the prediction effect of the model to ensure certain reliability? Can it be applied to short, medium and long term forecast?*

**Reply:** Thanks for the comments. Our method's implementation relies on relatively accurate machine learning prediction results, and therefore, we believe that the prediction time span is primarily constrained by the accuracy and duration of machine learning's future predictions. As we had previously tested a prediction horizon of up to 24 hours with success, we also tested the same prediction horizon in this work. The notation is in page 17, line 23-25: "***Note that our fused prediction skill also generally declines with an increase in prediction length following the RFSML. Therefore, prediction fusion with***

*a longer forecasting horizon was not carried out.*" Based on our experience, we have observed that the predictive performance of hourly PM$_{2.5}$ prediction tends to deteriorate rapidly when predicting beyond a 24-hour horizon. We have noticed that some studies involve daily predictions of atmospheric pollutants, and we are of the opinion that our methods could be adapted for such applications. To summarize, we believe that our method can be applied to short-, medium-, and long-term forecasts provided that we have the corresponding predictions from machine learning.

*What are the main reasons for the different prediction errors in different regions in Figure 9? For example, why does the prediction effect of PRD region have greater error and smaller R than that of other regions?*

**Reply:** Similar to other assimilation systems, the proposed method fundamentally relies on the Bayesian theory to determine the optimal posterior that aligns with the two priors quantified by their respective covariance matrices. In our case, these priors are the RFSML and CTM predictions, along with their corresponding covariance matrices. Figure 9 shows that the CTM prediction of PRD (black star) has greater error and a smaller R value. The similar results found in Figure 8 of our previous work (Fang et al., 2022), where the RFSML prediction of PRD had the smallest R value. We believe that both factors contributed to the greater error and smaller R value observed in PRD. Remarks about this plot will be added in page 17, line 11-12 by saying: "***The smallest R value of PRD can be attributed to both the smallest R value of RFSML and CTM predictions.***"

*What is the role of the integrated Kalman filter (EnKF) based on Bayesian theory?*

**Reply:** The Ensemble Kalman Filter is an extension of the standard Kalman filter that uses an ensemble of model simulations to represent the system state and its uncertainty, and the implementation follows the Bayesian theory. To explain this, extra description is added in page 5, line 1-4 "***The specific sequential assimilation system that is used to combine the site-available RFSML prediction and CTM prediction is the EnKF that was originally proposed by Evensen (1994) and further corrected by (Evensen, 2004). Similar to other assimilation algorithms, this assimilation system fundamentally relies on the Bayesian theory for finding the optimal posterior that fits the two priors quantified by their covariance matrices (Evensen et al., 2022).***"

*Does Figure 2 show that the prediction effect of fusion is worse than that of RFMSL according to actual observation.*

**Reply:** We agree with the referee that our prediction fusion might not perform as good as the RFSML over the site where RFSML prediction is available. However, the focus of our work is to predict at grids where RFSML prediction is not available, hence the significance of fusion prediction. Because the

RFSML prediction is only site specific as explained in page 3, line 23-28: "***We trained these models using historical measurement datasets collected at independent air quality monitoring stations, and they operate using real-time air quality observations as inputs. However, unlike gridded forecasts, our predictions are only available for the location of the air quality monitoring sites. Meanwhile, the spatial distribution of existing environmental monitoring stations is rather uneven in China, with a dense monitoring network in the east and a sparse network in the west, as shown in Fig.1. Therefore, our RFSML predictions limited to these few monitoring stations cannot accurately represent the true PM$_{2.5}$ concentrations on a national scale.***"

*It is suggested to add the advantages, disadvantages and prospects of the article.*

**Reply:** We appreciate your suggestions and agree that adding a section on the advantages, disadvantages, and prospects of the article would enhance the analysis of the topic. We now added a new paragraph in the conclusion that will provide an in-depth analysis of the advantages, disadvantages, and prospects of the topic. To explain this, remarks are now added in page 19, line 26-30 "***In summary, the proposed fused prediction effectively overcomes the weakness of machine learning, which can only predict at specific sites. However, our method has some drawbacks, such as 32 ensemble CTM forecasts which are still computationally expensive. Additionally, the site-based RFSML prediction may have unavoidable errors in representing atmospheric dynamics of the grid mean, which we will address in our future work. This method can be extended to predict the concentrations of other airborne pollutants.***"

*What is the effect if the model is applied to other seasons?*

**Reply:** Thanks for the comment. We hold the belief that our fusion prediction is capable of handling other seasons. As demonstrated above, the prediction fusion is determined by the RFSML and CTM predictions, along with their corresponding covariance matrices. In this study, we only showcased highly polluted seasons. To further demonstrate the robustness of the proposed method, we will also conduct testing for a less polluted month (April 2020). The overall performance is illustrated in the Taylor diagram, as presented in Supplementary Figure S6. Remarks about this plot will be added in page 17, line 27-30 by saying: "***To further showcase the robustness of the proposed prediction fusion approach, we conducted testing for a less polluted month (April 2020) using prediction horizons of 6 hours and 18 hours. The overall performance of each region is illustrated in Supplementary Figure S6 using a Taylor Diagram. Our results demonstrate that the prediction fusion method outperforms CTM in all regions, and the incorporation of covariance inflation further enhances this advantage.***"

[Figure]

***Figure S6. A modified Taylor Diagram that illustrates RMSE and R together. The regions of interest, including NCP, PRD, SCB, YRD, and FWP, are differentiated by unique markers of circle, star, square, triangle down, plus, and diamond, respectively. Additionally, the results from different approaches are visualized by distinct colors, with black, blue, and red indicating the results from CTM, prediction fusion without covariance inflation, and prediction fusion with covariance inflation, respectively. The diagram consists of two panels, representing forecasts for 6-hour and 18-hour time horizons, arranged in a left-to-right sequence.***

*You'd better analyze the prediction effect in different regions. Which regions are better predicted?*

**Reply:** We have discussed the performance of our method in different regions in page 17, line 18-22, by saying: "***For example, in panel a, the CTM has the worst prediction in terms of R (<0.1) in the PRD region, but it increases to 0.62 when the RFSML forecast is assimilated and further increases to 0.85 when covariance inflation is implemented. In terms of the RMSE, the most remarkable improvement is obtained in SCB region. Our EnKF-based fusion reduced the RMSE from 43 µg·m⁻³ to 12.73 µg·m⁻³ and 10.97 µg·m⁻³ (with covariance inflation).***". Remarks about the prediction performance for each region will be added in page 17, line 10-16 by saying: "***In terms of the Pearson correlation coefficient, SCB shows the best predictability among the regions, while PRD has the poorest performance. However, no significant differences were observed in the predictive performance of these regions. Regarding the Root Mean Squared Error (RMSE), NCP and FWP exhibited the largest values. This outcome can be attributed to the fact that NCP and FWP are located in the northern region of China, where the frequency of pollution days is higher due to adverse meteorological conditions and high emissions during winter. It should be noted that the metric RMSE is directly influenced by the atmospheric pollution levels, wherein higher PM₂.₅ concentrations tend to yield larger RMSE.***"

*In the introduction and other section, it is suggested to add some references on the application of machine learning and prediction models (such as: [https://doi.org/10.1016/j.scitotenv.2022.160928](https://doi.org/10.1016/j.scitotenv.2022.160928)).*

**Reply:** The suggested reference is indeed relevant and benefit our article. We now included the reference in the appropriate section of Introduction and ensure that it is properly cited in the text and included in the reference list. Remarks are now added in page 2, line 11-12 "***For instance, Chen et al. (2023) fully estimates hourly near-surface ozone concentration in China using a new geostationary satellite with the help of machine learning.***"

---

## Author Comment (AC2)

**Response to Referee #2:** We would like to thank the editor for the careful review throughout the paper that helps to improve our paper.

Our Reply follows (*the editor's comments are in italics and blue*)

*General Comments*

*This manuscript develops an EnKF-fusion method combining RFSML prediction and chemical transport model outputs to improve air quality forecasts, which has some good applications. However, the manuscript needs some major revision for publication. First, English languages need to be improved overall. Some of the sentences and words are causing confusion. Also, the organization of the manuscript can be improved to be more concise and clear. Please see the specific comments below.*

**Reply:** Thanks for the comments. We have polished the English languages thoroughly and reorganize the manuscript accordingly.

*Specific comments*

*1. The title is kind of confusing, it doesnot show the purpose of the work, yes, it is a method, but what it is used for?*

**Reply:** The title was changed to comply with the journal policies according to the editor's suggestion. We now changed it from "***EnKF-based fusion of site-available machine learning air quality predictions from RFSML v1.0 and gridded chemical transport model forecasts from GEOS-Chem v13.1.0***" to "***A gridded forecast through fusing site-available machine learning air quality predictions from RFSML v1.0 and chemical transport model forecasts from GEOS-Chem v13.1.0 using EnKF***" to make the purpose of manuscript more clearly.

*2. The abstract needs to be revised. It is not clear what the method used is and what the improvement is like from this method. Also, there is no need to discuss a lot of the technical details in the abstract. It should also focus on the significance of this work.*

**Reply:** To highlight the significance of the manuscript, **ABSTRACT** is now revised to "***Statistical methods, particularly machine learning models, have gained significant popularity in air quality predictions. These prediction models are commonly trained using the historical measurement datasets independently collected at the environmental monitoring stations, and their operational forecasts onward by the inputs of the real-time ambient pollutant observations. Therefore, these high-quality machine learning models only provide site-available predictions and cannot be used as the operational forecast solely. In contrast, deterministic chemical transport models (CTM), which simulate the full***

*life cycles of air pollutants, provide forecasts that are continuous in the 3D field. Despite their benefits, CTM forecasts are typically biased particularly on a fine scale owing to the complex error sources due to the emission, transport, and removal of pollutants. In this study, we proposed a fusion of site-available machine learning prediction, which is from our RFSML v1.0, and a CTM forecast. Compared to the normal pure machine learning model, the fusion system provides a gridded prediction with relatively high accuracy. The prediction fusion was conducted using the Bayesian theory-based ensemble Kalman filter (EnKF). Background error covariance was an essential part in the assimilation process. Ensemble CTM predictions driven by the perturbed emission inventories were initially used for representing their spatial covariance statistics, which could resolve the main part of the CTM error. In addition, a covariance inflation algorithm was designed to amplify the ensemble perturbations to account for other model errors next to the uncertainty in emission inputs. Model evaluation tests were conducted based on independent measurements. Our EnKF-based prediction fusion presented superior performance compared to the pure CTM. Moreover, covariance inflation further enhanced the fused prediction particularly in the cases of severe underestimation.*"

*3. P2, Line 1-2: the atmospheric environment has been improved, you mean air quality has been improved?*

**Reply:** Accepted. "***atmospheric environment***" is now changed to "***air quality***".

*4. P2, Line 3: how do you know if it is primary PM5 or secondary PM2.5?*

**Reply:** We meant the primary pollutant $PM_{2.5}$. To make it clear, "*primary*" is now removed.

*5. P2, Line 13: what do you mean by typical cities?*

**Reply:** Here typical cities mean megacities in China, including Shanghai and Chengdu. "***Cheng et al. (2021c) successfully predicted ground daily maximum 8-h ozone concentrations in typical cities of China by utilizing wavelet decomposition and two machine learning models.***" is now replaced with "***Cheng et al. (2021c) successfully predicted ground-level daily maximum 8-hour ozone concentrations in several megacities in China, Shanghai and Chengdu, by utilizing wavelet decomposition and two machine learning models.***" in page 2, line 16-18.

*6.P2, Line 15-25: These few sentences are quite confusing, I am not sure what the purpose is. Are you trying to say the RFSML is good or not?*

**Reply:** Sorry for the confusion. Our intention was to demonstrate both the advantages and disadvantages of the RFSML. Our prediction fusion is specifically designed to address the unavoidable disadvantages of the RFSML. Now we have reorganized these few sentences by replacing "******

*performed air quality prediction with high accuracy using the regional feature selection-based machine learning model (RFSML). Our forecast system is capable of providing national-scale short-term prediction over 1262 sites in China. Moreover, ensemble-SAGE feature selection algorithm was developed that can exclude those redundant inputs and efficiently improve the forecasting ability (Fang et al., 2022). These models were trained via the historical measurement datasets collected at the air quality monitoring stations independently, and their operational forecasts forwards with the inputs of the real-time air quality observations. Unlike the gridded forecast, these predictions are only available over the location of the air quality monitoring sites. Meanwhile, the spatial distribution of existing environmental monitoring stations is rather uneven. For example, the monitoring network in China is dense in east and very sparse in west as shown in Fig. 1*" with "*We have successfully used the regional feature selection-based machine learning model (RFSML) to predict air quality with high accuracy. Our forecast system can provide short-term predictions for over 1262 sites across China at a national scale. We developed the ensemble-SAGE feature selection algorithm to exclude redundant inputs, which efficiently improves our forecasting ability (Fang et al., 2022). We trained these models using historical measurement datasets collected at independent air quality monitoring stations, and they operate using real-time air quality observations as inputs. However, unlike gridded forecasts, our predictions are only available for the location of the air quality monitoring sites. Meanwhile, the spatial distribution of existing environmental monitoring stations is rather uneven in China, with a dense monitoring network in the east and a sparse network in the west, as shown in Fig. 1. Therefore, our RFSML predictions limited to these few monitoring stations cannot accurately represent the PM$_{2.5}$ concentrations on a national scale.*"

*7.Paragraph 3 and 4 can be combined to discuss why model forecasts and data assimilation are needed. For example, line 30-35 on P2 mentioned a lot of previous modeling work but did not mention how well the models did, which is the point.*

**Reply:** Thanks for the comment. The Paragraph 3 and 4 are of course closely related. The reason for separating them are as following:

(a) We would like to emphasis that air quality forecasts from CTMs are not perfect and list the main error sources in paragraph 3. Conclusive descriptions for the model forecast are added in page 3, line 4-9 by saying "***While those air quality forecast models can capture the spatiotemporal variations in ambient pollutants to some extent, they are susceptible to systematic bias, particularly at a fine scale, due to multi-source uncertainties in emission inventories (Keenan et al., 2009; Fan et al., 2018), initial and boundary conditions, and parameterization of physical and chemical processes such as transport and removal (Croft et al., 2012; Solazzo et al., 2017). This makes the CTM forecast less reliable for localized air quality predictions (Bi et al., 2022).***"

(b) In paragraph 4, we aimed to illustrate the both the CTM models and the observations have weakness when they are used in the reanalysis product. Data assimilation could combine them together and resulted in a gridded reanalysis dataset with high accuracy.

*8.P3, line 5: what are the same challenges here?*

**Reply:** More descriptions are now added in page 3 line 10-12 by saying "***Both the machine learning and CTM models have weakness when they are solely used in operational air quality forecasting. The same challenges exist when observations and simulation models are used to describe the atmospheric dynamics in reanalysis products.***"

*9.P3, line 17-26: this paragraph is not clear. I cannot tell what the innovation is and what exactly the method is.*

**Reply:** Thanks for the comment. We now rewrite this paragraph in page 3 line 24-35 as "***This study introduces the Bayesian theory-based assimilation method to fuse the regional-feature-selection machine learning forecast (RFSML v1.0) (Fang et al., 2022) and the deterministic chemical transport model (CTM) air quality prediction. The prediction fusion aims to achieve a gridded prediction with less bias and higher accuracy than the pure CTM prediction. It is continuous in 3D field unlike the machine learning forecast that is only site-available. To the best of our knowledge, this is the first time that the assimilation method has been applied in this way, as it is typically used for nudging model simulations with observations. The specific assimilation algorithm used is the ensemble Kalman filter (EnKF). The background error covariance of the CTM prior forecast is the fundamental term in the assimilation-based fusion. Ensemble CTM forecasts, which are driven by perturbed emission inventories, are forwarded in parallel to represent the potential distribution of ambient pollutant levels and the spatial covariance statistics. To avoid model divergence, an additional covariance inflation algorithm is developed that accounts for model errors other than uncertainties in emission inputs. The uncertainty of the other prior, the machine learning forecast, is also an essential part of the assimilation-based fusion. To accurately quantify the errors, dynamic covariance is designed.***"

*10.P4, line 1-5: this paragraph is repetitive from the last paragraph of the introduction. Instead, you can briefly mention the study domain and period here to give an overview.*

**Reply:** This paragraph is a duplicate and we have removed it accordingly. The study domain and period are introduced in ***Section 2.1 Study domain and observations***.

*11.P4, line 10: six categories, you mean 6 regions?*

**Reply:** The six categories refer to six regions, and we now use '*six regions*' to avoid any potential misunderstanding.

*12.P4, line 15: the winter of 2019, can you give the specific study period instead?*

**Reply:** It should be explicitly mentioned in this work. To explain this, remarks are now added in page 4, line 18-20 "***The winter of 2019 (from 15 October to 30 December 2019) was selected as the test period following the choice in our recently work (Li et al., 2022) as winter suffers the most severe haze pollution than other seasons in China.***"

*13.This work has tested the predication skill of the method for 6, 12, 18, 24h, would it work for longer time periods such 48h or more?*

**Reply:** We aimed to combine the accurate short-term machine learning forecast and the CTM forecast. The fused prediction then relied on the machine learning prediction results. As we had previously tested a prediction horizon of up to 24 hours with success, we also tested the same prediction horizon in this work. Based on our experience, it was observed that the RFSML predictive performance tends to deteriorate rapidly when predicting beyond a 24-hour horizon. Hence, we do not recommend using RFSML for predicting longer time periods. The notation is in page 17, line 23-25: "***Note that our fused prediction skill generally declines with an increase in prediction length following the RFSML. Therefore, prediction fusion with a longer forecasting horizon (>24 h) was not carried out.***"

*14.I am not sure if Figure 2 can really explain much of the fusion method. For section 2.3, it is not clear how the RFSML prediction works, for example, what are the inputs used? Also, how do you first acquire the RFSML prediction and then do the fusion?*

**Reply:** Thanks for the comments. We agree that the inputs and their relationships would make the schematic plot more comprehensible. We have modified the Figure 2 as shown below.

[Figure]

***Figure 2. Framework of EnKF-based prediction fusion. The blue lines and their shadows imply for the RFSML prediction and its uncertainty at air quality monitoring stations, which are very close to the PM2.5 reality. The gold surface and its surrounding grey surfaces stand for the CTM forecast and its uncertainty. The medium slate blue surface represents the fusion prediction of RFSML and CTM forecast.***

The RFSML prediction is acquired from our last work "***Development of a regional feature selection-based machine learning system (RFSML v1.0) for air pollution forecasting over China***" (Fang et al., 2022). We will clearly explain it in ***Section 2.3 RFSML prediction & uncertainty*** in page 8, line 1-2 by saying "***Our RFSML is capable of providing the operational air quality prediction with a maximum horizon of 24 h. The RFSML prediction results used in this study are directly acquired from our last work (RFSML v1.0).***"

*15.Figure 4, time series of surface PM5, right? This should be added in the caption. Also, where are the ground observations on the plot?*

**Reply:** The mistake in the title is now corrected as shown below. Additionally, we have identified similar mistakes in the titles of Figure 4, Figure 6, and Figure S4, which are also corrected accordingly. Figure 4 was to demonstrate that the posterior forecast error sometimes remained high, particularly when the prior chemical transport model (CTM) significantly underestimated pollution levels. Mathmatically, this is because the CTM spread (shown as a silver shadow) was much lower than the uncertainty estimated by the random forest statistical machine learning (RFSML) model (shown as a blue shadow). Therefore, we did not include the ground observation here.

[Figure]

***Figure 4. Panels a and b are the time series of an environmental monitoring station (Latitude:40.98°N, Longitude:117.95°E) in Chengde, Hebei Province. RFSML prediction is available at this station and will be assimilated. Markers of medium blue solid line, deep sky-blue solid line, dark violet solid line, and black dot represent the RFSML prediction, prediction fusion without covariance inflation, prediction fusion with covariance inflation, and mean of ensemble CTM predictions respectively. The***

*silver and blue shadows represent the uncertainty of the CTM and RFSML respectively. Panel c is the inflation function.*

*16.For Figure 6, can you also use scatter plots or Taylor diagrams to show the overall performance compared with ground observations?*

**Reply:** Thanks for the suggestion. Scatter plots are now added to highlight the superior performance more clearly. Remarks about the plot will be added in page 13, line 26-30 by saying: "***To further highlight the superior forecast skill, we present the distributions of both prediction fusion methods and ground observations for a 6-hour prediction horizon in supplementary Figure S5. The results clearly indicate that the prediction fusions align closely with the ground observations, and that the prediction fusion with covariance inflation effectively addresses the underestimation issue present in the prediction fusion without covariance inflation.***"

[Figure]

***Figure S5. The distributions of model predictions and observations for a 6-hour predicting horizon. Panels (a), (b), and (c) illustrate the distributions between CTM, prediction fusion without covariance inflation, and prediction fusion with covariance inflation, and their corresponding ground observations, respectively.***

*17.Section 2.5: the radius tested is 3°, 5° and 10°. The distance is quite far, even using 3°, the distance between a ground station and grid would be about 300 km, the correlation between these two places would be low, if I understand the method correctly. If you reduce the radius, I assume you can see some impacts in the city area where ground observations are clustered. There is still value using the interpolation method. Also, what would be the computational cost for the interpolation method vs the fusion method? It is not quite convincing here just to make this comparison and I think it depends on the application.*

**Reply:** Reducing the radius of spatial interpolation of course can improve its performance, especially in areas with dense ground observations. However, our work was primarily focused on addressing the challenge of producing gridded predictions across China. Cressman interpolation with radius less than

3° could meet the requirement. Therefore, we believe it is suitable for this specific task. So, we did not discuss the computational cost of the interpolation method vs the fusion method and just passed the interpolation method. We agree with the referee that it is worth noting that the interpolation method is computationally less expensive. In contrast, the fusion method requires running ensemble CTMs, which is much more computationally intensive. We now made these points more clearly by saying: "***It is worth noting that the interpolation method is computationally less expensive than the fusion method, and it can be a powerful tool for gridded prediction when there are plenty of ground observations available.***" in page 12, line 13-15.

*18. What is the computational cost of this fusion method? What is the application of this method? Some discussions are needed for the implication of this work.*

**Reply:** A new section for describing the computational complexity in ***Section 3.4*** by saying: "***In this study, all computations related to the prediction fusion were carried out on nodes equipped with 4 x 16-core 2.1 GHz Intel Xeon E5-2620 v4 CPUs and with memory 64 GB. The RFSML was demonstrated to be relatively efficient in computation as illustrated in Fang et al., (2022). While the ensemble CTM forecasts takes the most computation power, which however could be implemented in parallel. Each CTM model takes approximately 30 minutes to run a 24-hour simulation on average, with only 16 cores. The computational cost for EnKF fusion is also low, with an average time of three minutes for a prediction fusion. Overall, the proposed prediction fusion is time affordable.***"

---

## Author Response (AR2)

**Response to Editor:** We would like to thank the editor for the careful review throughout the paper that help to improve our paper.

Our Reply follows (*the editor's comments are in italics and blue*)

*Thank you for revising the manuscript. I would like to ask you to address the remaining comments of referee 2 (quoted below) and consider the following in a minor revision:*

*When discussing different forecast ranges, please make it clearer that the CTM results used are always the same and are based on reanalysis rather than forecasts (even if true CTM forecasts would be used in future applications). Accordingly, please avoid the term "CTM forecast" where it is not appropriate, e.g., in Figs. 4, 6, 7. For example, you could use "CTM ensemble", "CTM prediction", or "CTM result".*

*Also in the title I suggest replacing "chemical transport model forecasts" by "chemical transport model results", and moreover placing "air quality" more prominently:*
*"A gridded air quality forecast through fusing site-available machine learning predictions from RFSML v1.0 and chemical transport model results from GEOS-Chem v13.1.0 using EnKF"*

**Reply:** Thanks for the suggestion. The term "***CTM forecast***" is changed into either "***CTM ensemble***" or "***CTM prediction***" depending on the context throughout the paper. More descriptions are now added in page 9 line 1 by saying "***Note that the CTM results utilized in this work remain consistent regardless of changes in the forecast horizon.***" All figures that originally displayed "***CTM forecast***", including Figs. 4, 6, and 7, have been updated to show either "***CTM ensemble***" or "***CTM prediction***". The title is now changed to "***A gridded air quality forecast through fusing site-available machine learning predictions from RFSML v1.0 and chemical transport model results from GEOS-Chem v13.1.0 using EnKF***".

*Minor comments*
*Page 7, line 19: "188 stations that failed" should probably be rephrased to "the 188 stations that where skipped".*

**Reply:** Accepted. The sentence in page 8, line 2-3 is changed to "***For this study, the 188 stations that were skipped in the RFSML model training were used for validating our fused prediction, they are referred to as validation sites B and marked as black rectangles in Fig. 1.***"

*Fig. 3: Please mention in the caption that the averages in panel b correspond to the intervals shown in panel a.*

**Reply:** Thanks for the suggestion. The remark "***The averages in panel b correspond to the intervals depicted in panel a.***" is now added to the end of the caption for ***Figure 3***.

*Fig. 5: The colour scale label is missing.*

**Reply:** The missing colour scale label is added now by "***PM$_{2.5}$ (µg/m$^3$)***".

*Fig. 9: In the panel titles, the units ("h") of the forecast ranges are missing.*

**Reply:** The units "***h***" is now added in ***Fig. 9*** and ***Figure. S6***.

**Response to Referee:** We would like to thank the referee for the careful review throughout the paper that helps to improve our paper.

Our Reply follows (*the referee's comments are in italics and blue*)

*General Comments*

*The manuscript has improved after the authors have addressed the previous comments. I still have a few specific comments and suggestions as follows:*

*Specific comments*

*1. Page 4: line 15: top three significant features, what are they?*

**Reply:** The context concerning the top three significant features for the RFSML prediction was not placed in the right place. They are now added to ***Section 2.3 RFSML prediction & uncertainty*** in page 7 line 18-20 for better organization and coherence. "***Using an ensemble SAGE selection, we identified the top three significant features for each region, as outlined in Supplement Table S1 (Table 6; Fang et al. (2022)).***"

**Table S1.** Summary of selected features.

| Region | NCP | PRD | SCB | YRD | FWP | REST |
|---|---|---|---|---|---|---|
| | $PM_{2.5}$ | $PM_{2.5}$ | $PM_{2.5}$ | $PM_{2.5}$ | $PM_{2.5}$ | $PM_{2.5}$ |
| **Feature** | v10 | v10 | d2m | v10 | d2m | co |
| | co | pm2p5 | tp | pm2p5 | co | pm2p5 |

*2. Page 4: line 16: preceding nine hours. It is unclear what the 9 hours refer to.*

**Reply:** Sorry for the confusion. The choice concerning the $t_p$ = 9h in ***Eq. 9*** in ***Section 2.3 RFSML prediction & uncertainty*** was determined based on the results of auto-correlation and partial auto-correlation. Remarks are now added in page 7, line 23-25, by saying: "***where at any instant t, the input vector storing s = 3 individual selected features over the previous $t_p$ = 9 h is utilized to forecast the target PM₂.₅ concentrations ŷ with a prediction horizon of h h. The choice of $t_p$ = 9 h is obtained on the basis of the auto-correlation and partial auto-correlation analysis.***"

*3. It is still not clear how the RFSML prediction works. Yes, it is published in previous paper, but Section 2 should give a brief introduction how the prediction is made. I would suggest updating Section 2.3 to introduce how RFSML works.*

**Reply:** Thanks for the comment. We provided a more specific introduction to RFSML, with relevant comments on the page 7, line 13-19 by saying: "***In addition to the following steps in machine learning***

*prediction, which include data collection of PM2.5 observations and datasets, interpolation of missing values in the original dataset, selection of an appropriate machine learning model, reformation of the continuous data time series into the required input structure, repeated training of the model to determine optimal hyperparameters, and making predictions using the trained model. The RFSML utilized ensemble SAGE to obtain the optimal input feature subsets. The total national air quality monitoring stations were divided into six regions. Using a computational efficient ensemble SAGE selection, we identified the top three significant features for each region, as outlined in Supplement Table S1 (Table 6; (Fang et al., 2022)).*"

*4. Figure 2 is still not clear to explain the EnKF framework. It is hard to follow the flow and the caption does not explain well either. It should be clearly showing what the inputs and outputs are for the system.*

**Reply:** In Figure 2, we have incorporated the inputs and outputs to improve the clarity and understandability of the framework. Additionally, we have included a more specific description in the caption to provide additional context.

[Figure]

**Figure 2. Framework of EnKF-based prediction fusion. The blue lines and their corresponding shaded regions represent the RFSML predictions and their uncertainty at the air quality monitoring stations, which are assumed to be very close to the actual $PM_{2.5}$ concentration values. The golden surface and its surrounding grey surfaces represent the CTM prediction and its uncertainty. The medium slate blue surface represents the fused prediction of the RFSML and CTM prediction. $y$ and $[x_1^f, ..., x_N^f]$ are the inputs of EnKF, which represent RFSML prediction and ensemble CTM respectively.**

*5. Figure S3: what does the color bar represent?*

**Reply:** The missing label "$PM_{2.5}(\mu g/m^3)$" has now been added to the Figure S3.

---

## Author Response (AR3)

**Response to Editor:** We are grateful to the editor for their diligent review of this article once again.

Our Reply follows (*the editor's comments are in italics and blue*)

*General Comments*

*Many thanks for the revised manuscript. There are just a few minor points that need additional attention:*

*Minor comments*

*The newly added short introduction of the RFSML is very welcome, but I would like to ask you to check and revise the corresponding text on page 7. The first new sentence, lines 13 to 17, appears to be grammatically incomplete and is also quite long. Moreover, important information such as the type of ML model used is missing.*

**Reply:** Thanks for the comments. The long sentence is revised by *"The common machine learning prediction process involves several steps. Firstly, it requires data collection of PM$_{2.5}$ observations and datasets. Next, data interpolation should be conducted to address missing values in the original dataset. Following that, an appropriate machine learning model must be selected. Additionally, the continuous data time series should be reformed into the required input structure. Then, the model is repeatedly trained to determine optimal hyperparameters. Finally, predictions can be made using the trained model. In addition to these procedures, the RFSML utilized ensemble SAGE to obtain the optimal input feature subsets instead of using all related features."* in page 7, lines 13-19. Information concerning the type of ML model is now added in page 7, line 26 and page 8, lines 1-2 by saying *"The forecast predictor F represents the machine learning model. In RFSML, three machine learning models, namely random forest, gradient boosting, and multi-layer perceptron (MLP), are employed. The prediction results obtained from MLP are directly utilized as this work's RFSML prediction."*

*Minor comments*

*Page 2, line 22: Please expand SAGE (in parenthesis).*

**Reply:** The full name (*Shapley additive global importance*) of SAGE has been added in parentheses.

*Page 13, lines 5, 6, 8: Consistent with the renaming elsewhere, please change "forecast" to "prediction".*

**Reply:** Corrected.